# Farmland Shelterbelt Changes in Soil Properties: Soil Depth-Location Dependency and General Pattern in Songnen Plain, Northeastern China

**Yan Wu [2], Wenjie Wang [1,3,\*], Qiong Wang [4], Zhaoliang Zhong [5], Huimei Wang [1] and Yanbo Yang [6]**

1   State Key Laboratory of Subtropical Silviculture, College of Forestry and Biotechnology, Zhejiang A&F University, Hangzhou 311300, China; 20230011@zafu.edu.cn
2   College of Biological Sciences, Guizhou Education University, Guiyang 550018, China; wu_yan@gznc.edu.cn
3   Northeast Institute of Geography and Agroecology, Changchun 130102, China
4   College of Forestry, Jiangxi Agricultural University, Nanchang 330045, China; wangqiong@jxau.edu.cn
5   College of Resources and Environment, Jiujiang University, Jiujiang 332005, China; zhongzhaoliang999@163.com
6   Key Laboratory of Forest Plant Ecology (MOE), College of Chemistry, Chemical Engineering and Resource Utilization, Northeast Forestry University, Harbin 150040, China; 15049065676@163.com
\*   Correspondence: wwj225@nefu.edu.cn

**Abstract:** As one of the world's largest ecological rehabilitation programs, the three-north (Northern China, Northeastern China, and Northwestern China) shelterbelts program in China were not well evaluated on its effects on multiple soil properties. This paper aims to quantify this. Seven hundred twenty soils from paired plots of farmlands and neighbor shelterbelts were sampled from six regions of Songnen Plain in northeastern China. Multivariate analysis of variance and regression analysis were used to detect the impacts of shelterbelt plantations. For the overall 1 m soil profiles, shelterbelt plantations had a 4.3% and 7.4% decreases in soil bulk density and soil moisture ($p = 0.000$), a 4.8% increase in soil porosity ($p = 0.003$). It also evidently recovered soil fertility with a 40% increase in total P, a 4.4% increase in total K, and a 15.1% increase in available K ($p < 0.05$). However, without overall changes were in SOC and N ($p > 0.05$). Compared with farmland, shelterbelt plantations produced a 7.8% SOC increase in 20–40 cm soil and much more minor changes in surface soil (0–20 cm). Compared with the younger plantation, mature shelterbelts tended to sequestrate more SOC in soils (from a 0.11% decrease to a 3.31% increase) and recover total K from a 2.24% decline to a 16.5% increase. Correlation analysis manifested that there is a significant relationship between SOC sequestration and the changes in bulk density, porosity, soil moisture, pH, EC, total N, total P, and alkaline hydrolyzed N. In contrast, the strongest relationship was observed between total N and SOC ($r > 0.50$, $p < 0.001$). The increase in total N was accompanied by 1.01–1.67-fold higher SOC sequestration in deep soils >20 cm in poplar forests. Our results highlight that the over-40-year shelterbelts afforestation on farmland in northeastern China could strongly affect soil physics, soil water, and nutrient of P and K. The effects on SOC sequestration were dependent on soil depths, growth stages, and regions. Our data support the precise soil evaluation of agroforestry projects in the black soil region in the high-latitude northern hemisphere.

**Keywords:** poplar shelterbelts; soil properties; carbon sequestration; multivariate analysis of variance; farmlands

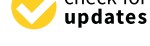



## 1. Introduction

The global soil carbon pool has been estimated to be over three times higher than the atmospheric pool and about four times as much as that in the biotic pool. Soil carbon pools are regarded as important atmospheric $CO_2$ sources or sinks, and any change in their reserves will vastly change $CO_2$ concentration and global carbon balance [1]. At present, afforestation in abandoned farmland has become essential in increasing C sinks in

many countries [2,3]. However, inconsistent results have been reported on the effects of afforestation on SOC, i.e., SOC accumulation [4,5], SOC loss [6,7], and initial-loss-then-SOC-gains [8,9]. Due to these contradictions, more studies are needed to evaluate the potential of SOC accumulation under afforestation in farmland and to find the dominant factors affecting SOC sequestration, which may favor soil management practices of plantation forests as well as farmland.

Afforestation in farmland soils could induce changes in most of the soil properties and soil fertility of bulk density, porosity, pH, EC, nitrogen, and other elements [5,10]. Fast-growing plantations, such as poplar, larch, or eucalyptus, consume more water and soil nutrients [11,12]. Soil improvements after forest plantation on degraded farmland have also been observed in different cases [13,14]. Black soils in northeastern China mainly locate in Songnen Plain and Sanjiang plain, one of the world's four black soil belts. Over 45% of northeast China's grain output is produced in this black soil region [15]. Historically, the black soil in northeastern China contains much soil organic matter (SOM) and is of high fertility compared with other soils [16,17]. Excessive reclamation has sharply reduced soil fertility since establishing the People's Republic of China in 1949 [18]. Near half of N and SOM have been lost in northeastern China [19,20]. Soil physical degradation has been found in soil bulk density increase, total porosity decrease, water retention, and ventilation capacity decrease [5]. Soil fertility and quality changes after afforestation on long-term degraded farmland are worthy of a detailed study for evaluating the national forest policy of the Three-North shelterbelts program [21].

The "Three-North Shelterbelts" program was launched in the Northwest, North China, and Northeast of China in 1978, mainly for farmland protection from severe erosions and windstorms. The project is called "Green Great Wall" and contains 551 counties of 13 provinces in China. The total area is 4.07 $Mkm^2$, accounting for 42.4% of the national territorial area [22]. In northeastern China, shelterbelt plantation has increased from less than 5% to over 15% of land area. As a primary shelterbelt type in this region, farmland shelterbelts have protected the soil against wind erosions and increased crop yields by improving the field microclimate environment [23] and soil matrix protection [24]. The importance of SOC sequestration, soil physics, and soil fertility from farmland shelterbelts construction have not been systematically studied yet [25,26]. Some basic questions still need to be answered. Can farmland shelterbelt construction improve soil properties and increase soil fertility in the whole 1 m soil profile? Does SOC sequestration potential differ from sites, soil depths, and tree growth stages? Additionally, which soil factors are mainly responsible for such variations? The answers to these questions are crucial for farmland shelterbelt practices regarding the local and global shelterbelt plantations management and soil maintenance both in the view of short-term and long-term.

Poplar plantations are about 6.67 Mha in China, and poplar is one of the main species used in farmland shelterbelt practices in northeastern China [22,27]. Poplar shelterbelts can affect wind velocity and adjust crop growth environment [28], change SOC and soil respiration [3], and soil microbial biomass and activity [25]. However, there is still considerable controversy on the size of these impacts [29]. The massive amount of poplar shelterbelts and the flat topography make it possible to compare shelterbelt influences on soil fertility, soil physical-chemical properties, and SOC sequestration with references to neighbor farmland in northeastern China.

In this paper, we hypothesize that poplar shelterbelts construction can improve multiple soil properties for recovery of soil fertility of the 1 m soil profile, and they strongly regulated the afforestation-induced SOC sequestration in different regions and forest age groups. To test the hypothesis, we compared the variation of SOC, soil fertility, and soil physical-chemical properties in different soil depths of shelterbelts and farmland to explore the soil improvements and correlations with SOC sequestration in this region.

## 2. Materials and Methods

### 2.1. Natural Condition of Study Sites and Soil Sampling Procedures

Songnen Plain, a total area of 18,300 hm², is located in the middle of northeastern China, involving 36 million arable lands. This region belongs to a continental monsoon climate, with an average precipitation of about 350–500 mm and an annual temperature of 2–4 °C. The 'Three-North' shelterbelts program is China's largest artificial ecological project, which constructs protection forests around existing farmland in 3-north regions [21]. A general design is to plant poplar shelterbelts 4–10 rows around 500 m × 500 m farmland. Moreover, shelterbelts were planted on the fixed site since its plantation in 1978. When plantations are mature enough for timber harvest, the new plantation should be planted in the same area again according to the regulation of this project. This kind of national shelterbelts afforestation policy provides a long-term fixed plot for our study. Farmers have dug out ditches of about 2 m in width and 2 m in depth between shelterbelts and farmland to decrease the influences on neighboring farmland productivity. They want to prevent the nutrient depletion of poplar roots from neighboring farmland, and such ditches have usually been there for dozens of years. This made this region an excellent place for paired sampling study. Maize was the main crop in farmland for at least five years, while the poplar plantation has been afforested since 1978.

Soil samples were collected from 72 paired plots of farmlands and shelterbelt plantations in 6 typical regions (Lanling, Zhaodong, Dumeng, Zhaozhou, Mingshui, Fuyu) uniformly distributed in Songnen Plain (Figure 1). One pair of soil profiles, approximately 1 m in length, 0.8 m in width, and 1 m in depth, was dug out in poplar shelterbelts and neighbor farmland, respectively. Twelve soil profiles of shelterbelts forests and 12 soil profiles of farmlands were dug out in each study region. Soil samples were collected from five soil layers (0–20 cm, 20–40 cm, 40–60 cm, 60–80 cm, and 80–100 cm) with a 100 cm³ cutting ring in each soil profile. In total, 720 soil samples (6 regions × 24 profiles × 5 depth = 720 samples) were collected. Basic information regarding sampling sites and soil types could be found in previous papers [10,14].

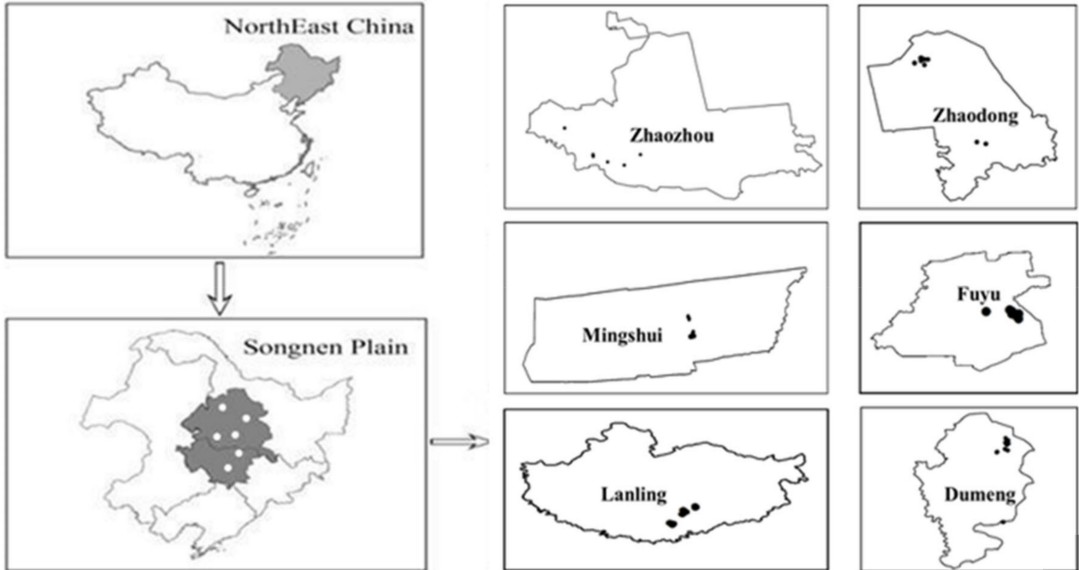

**Figure 1.** Six study sites in Songnen Plain, Northeastern China. Left: the relative location of the study sites in China and in Songnen Plain. Right: the relative location of the sites in six regions (Zhaozhou, Zhaodong, Mingshui, Fuyu, Lanling and Dumeng).

*2.2. Sample Preparation and Determination of Specific Gravity, Soil Bulk Density, Porosity, Soil Moisture, pH, EC, SOC, N, P, K*

The dried soil samples were ground and passed through a 2 mm sieve. The soil sample of the <2 mm component was smashed for approximately 3 min and passed through a 0.25 mm sieve. The 0.25 mm soil samples were collected for experimental analysis.

A pycnometer was used to determine the soil-specific gravity. The soil bulk density = the weight of air-dried soil/400 cm$^3$. Soil porosity and soil moisture were calculated with the following formulas:

$$\text{Soil porosity} = (1 - \text{bulk density}/\text{specific gravity}) \times 100\%$$

$$\text{Soil moisture} = (\text{fresh weight} - \text{dry weight})/\text{dry weight} \times 100\%$$

The pH of the soil solution was measured with a pH meter, and soil electrical conductivity (EC) was determined with an EC meter [30].

SOC was measured using the heated dichromate/titration method. Total soil nitrogen concentration was determined by the Semimicro-Kjeldahl method. Soil alkaline hydrolyzed N was determined with alkaline hydrolysis diffusion method. Total soil P and total K concentration were measured using sodium hydroxide melt method. Soil's available P content was determined by sodium hydroxide melting and molybdenum antimony colorimetric method. Soil K content was determined with flame photometry. All the procedures were from Bao [30], and some detailed descriptions can be found in Wang, Qiu [5] and Wu, Wang [14].

*2.3. Data Analysis*

To examine the influences of shelterbelts construction on SOC, soil fertility, and soil physical-chemical properties, as well as possible differences with soil depths, regions, and tree growth, a multivariate analysis of variance (MANOVA) was used to find differences in 15 soil parameters. Five independent factors include two land uses (farmland, shelterbelts), six sampling regions (Lanling, Zhaodong, Dumeng, Zhaozhou, Mingshui, Fuyu), five soil depths (0–20 cm, 20–40 cm, 40–60 cm, 60–80 cm, 80–100 cm), three tree height groups (<12 m, 12–18 m and >18 m), 3 DBH groups (<15 cm, 15–30 cm, and >30 cm). The 14 soil parameters include soil carbon (SOC concentration, SOC stock), soil fertility (total N, alkaline hydrolyzed N, total K, available K, total P, available P), and soil physical-chemical properties (specific gravity, bulk density, porosity, soil moisture, pH, EC). SPSS 17.0 was used in this analysis.

SOC sequestration potential was described as the differences between shelterbelt and farmland (shelterbelt-farmland) in SOC concentration and storage. The storage was calculated with the following formula:

$$\text{SOC storage} = \alpha \times \rho \times H_i \times \left(1 - V_{\text{gravel}}\right) \times a$$

where $\alpha$ represents the concentration of SOC (g kg$^{-1}$), $\rho$ represents soil bulk density (g cm$^{-3}$), $H_i$ represents soil thickness (m) of i layer, and $V_{\text{gravel}}$ represents the volume percentage of gravel. Correlation analysis between shelterbelt-induced SOC changes and soil physical-chemical or fertility parameters changes (specific gravity, bulk density, porosity, soil moisture, pH, EC, total N, alkaline hydrolyzed N, total K, available K, total P, available P) changes was conducted by using JMP 5.0.1.

## 3. Results

*3.1. Shelterbelt Plantations Construction Influences on Various Soil Parameters and Possible Interaction with Sites, Soil Depths, and Tree Growth: MANOVA*

For overall data (different sites and whole 1 m soil profile), shelterbelts did not change SOC concentration and stock ($p > 0.05$). Significant impacts were observed in soil fertility

parameters (total K, total P, available K) and soil physical parameters of soil bulk density, porosity, and soil moisture ($p < 0.05$) (Table 1).

**Table 1.** Effect shelterbelts construction on SOC, physical-chemical and fertility parameters, and possible interaction effect from sites, soil depths, and tree growth status.

| Dependent Variable | Type | | Type×Site | | Type×Depth | | Type×H | | Type×DBH | |
|---|---|---|---|---|---|---|---|---|---|---|
| | F | *p*-Value | F | *p*-Value | F | *p*-Value | F | *p*-Value | F | *p*-Value |
| Soil C parameters | | | | | | | | | | |
| SOC concentration(g/kg) | 0.569 | 0.451 | *29.133* | *0.000* | *125.718* | *0.000* | *2.674* | *0.031* | 0.964 | 0.426 |
| SOC stock(kg/m²) | 1.079 | 0.299 | *27.385* | *0.000* | *125.913* | *0.000* | 1.880 | 0.112 | 1.133 | 0.340 |
| Soil physical-chemical parameters | | | | | | | | | | |
| Specific gravity | 3.694 | 0.055 | 1.581 | 0.108 | *2.467* | *0.012* | 0.410 | 0.801 | 0.683 | 0.604 |
| Bulk density (g/cm³) | *86.835* | *0.000* | *31.497* | *0.000* | *8.872* | *0.000* | *4.070* | *0.003* | 0.740 | 0.565 |
| Porosity (%) | *8.833* | *0.003* | *6.745* | *0.000* | *7.268* | *0.000* | *0.862* | 0.487 | 1.171 | 0.322 |
| Soil moisture(%) | *13.676* | *0.000* | *105.258* | *0.000* | *10.634* | *0.000* | *6.887* | *0.000* | 0.308 | 0.873 |
| pH | 3.004 | 0.083 | *60.979* | *0.000* | *10.537* | *0.000* | *5.595* | *0.000* | *3.972* | *0.003* |
| EC(uS/cm) | 0.319 | 0.572 | *8.132* | *0.000* | *7.483* | *0.000* | 0.923 | 0.450 | 0.842 | 0.499 |
| Soil fertility parameters | | | | | | | | | | |
| Total N (g/kg) | 0.230 | 0.632 | *14.865* | *0.000* | *81.900* | *0.000* | 2.031 | 0.088 | 1.649 | 0.160 |
| Alkaline hydrolyzed N (mg/kg) | 0.158 | 0.691 | *5.077* | *0.000* | *41.211* | *0.000* | 0.721 | 0.578 | 0.328 | 0.859 |
| Total K (g/kg) | *4.438* | *0.036* | *6.108* | *0.000* | *22.811* | *0.000* | 1.465 | 0.211 | *2.456* | *0.045* |
| Available K(mg/kg) | *6.249* | *0.013* | *6.493* | *0.000* | *21.999* | *0.000* | 0.500 | 0.736 | 0.504 | 0.733 |
| Total P (g/kg) | *9.284* | *0.002* | *4.410* | *0.000* | *11.336* | *0.000* | 0.411 | 0.801 | 0.198 | 0.940 |
| Available P (mg/kg) | 1.016 | 0.314 | *15.556* | *0.000* | *8.854* | *0.000* | 1.681 | 0.153 | 2.047 | 0.086 |

Note: Bold, italic figures present a significant ($p < 0.05$) or extremely significant ($p < 0.01$) effect or interaction in the table.

The influences of shelterbelt plantations construction on soil parameters were dependent on sampling sites, soil depth, tree height, and DBH, as shown by the significant interactions in Table 1. For example, the effects of shelterbelt plantations on the following 13 parameters of SOC concentration: SOC stock, bulk density, porosity, soil moisture, pH, EC, total N, alkaline hydrolyzed N, total K, available K, total P, and available P were significantly interacted with sites (type*site), indicating that these shelterbelt-induced changes significantly differed among 6 sites ($p = 0.000$); the 14 parameters of SOC concentration, SOC stock, specific gravity, bulk density, porosity, soil moisture, pH, EC, total N, alkaline hydrolyzed N, total K, available K, total P, and available P existed interaction between land use type and soil depth (type*depth), showing that these shelterbelt-induced changes were different in 5 soil depths ($p = 0.000$). There was interaction between land use type and tree height group (type*H), too, indicating that shelterbelt-induced change of SOC concentration, soil bulk density, soil moisture, and soil pH differed at different tree height group ($p < 0.05$). Similarly, the interaction type*DBH indicates that shelterbelt-induced change of total K and pH differed at different DBH group.

*3.2. Overall Influences of Shelterbelts Construction on Soil: Parameters, Size, and Pattern of the Influences*

As a further step of the MANOVA, marginal means of different types of vegetation (farmland and poplar shelterbelts) with significant differences ($p < 0.05$; Table 1) were estimated as shown in Table 2, while those without differences were not listed here. Table 2 showed that shelterbelts construction resulted in a 4.3% decreased soil bulk density ($p = 0.000$), a 4.8% increased soil porosity ($p = 0.003$), a 7.4% decreased soil moisture ($p = 0.000$), a 40% increased total P ($p = 0.002$), a 4.4% increased total K ($p = 0.036$), and a 15.1% increased available K ($p = 0.013$). Overall data analysis for all other soil parame-

ters showed that the differences between shelterbelts and farmland were not statistically significant ($p > 0.05$).

**Table 2.** Shelterbelts construction significantly affected 6 soil parameters ($p < 0.05$), and magnitudes and pattern of the influences. The non-significant factors ($p > 0.05$) were not listed here.

| Type | Bulk Density g/cm$^3$ | Porosity % | Soil Moisture % | Total K g/kg | Available K mg/kg | Total P g/kg |
|---|---|---|---|---|---|---|
| Farmland | 1.472 | 38.865 | 12.443 | 50.760 | 62.346 | 0.340 |
| Poplar | 1.408 | 40.733 | 11.522 | 53.010 | 71.784 | 0.476 |
| Change (%) | −4.3 | 4.8 | −7.4 | 4.4 | 15.1 | 40.0 |
| *p*-value | 0.000 | 0.003 | 0.000 | 0.036 | 0.013 | 0.002 |

*3.3. Shelterbelt-Induced Soil Changes Dependence on Sampling Sites: Parameters, Size, and Pattern of the Changes*

The significant interaction between land use type and sites (type*site) was mainly in 13 soil parameters (Table 3), indicating that the difference between shelterbelts and farmland differed among 6 regions (Table 1, $p = 0.000$). Table 3 lists specific sizes and patterns of the differences.

Differences in SOC concentration between farmland and shelterbelt plantations varied among sites. Plus, values (accumulation of SOC according to farmland) were observed in Dumeng, Fuyu, Lanling, and Zhaodong. Negative values (depletions of SOC with references to farmland) were found in Mingshui and Zhaozhou. The range of SOC changes compared with farmland in 6 sites was −4.8% to 9.8%. Different from SOC concentration, the accumulations of SOC stock were found in Lanling and Zhaodong, while the depletions were discovered in Dumeng, Fuyu, Mingshui, and Zhaozhou. The range was from a 10.4% decrease to a 7.6% increase in SOC stock compared with those in farmland.

In the case of soil physical-chemical properties, type*site interactions were found in bulk density, porosity, soil moisture, pH, and EC (Table 3). Although an overall 4.3% decrease in bulk density was found in shelterbelt plantations (Table 2), a wide range of bulk density changes (0.7% decrease to 8.8% decrease) in shelterbelts was observed. It differed from different regions ($p = 0.000$). An 8.9% decrease in soil porosity of shelterbelt plantations compared to farmland was found in Zhaodong, and all other sites showed 1.0%–14.8% increases (Table 3). For the whole data average, the soil porosity of shelterbelt plantations increased by 4.8% compared to that of farmland (Table 2). Soil moisture reductions were found in all six regions, ranging from a 26.2% decrease in Dumeng to a 2.3% decrease in Fuyu. The type*site interactions indicate that the differences were statistically significant among different regions (Table 3). Compared with farmland, 0.1%–1.8% increases (on average 0.9%) in soil pH in shelterbelt plantations were found (Table 3). The change in EC of shelterbelt plantations compared to farmland ranged from a 25% decrease to a 20% increase with an average of 2.5% decrease, and shelterbelt-induced EC changes differed from different regions ($p = 0.000$).

**Table 3.** Shelterbelt construction and sampling sites significantly affected 13 soil parameters (marked type*site interaction, $p < 0.05$) and the magnitudes and pattern of the influences. The non-significant factors ($p > 0.05$) were not listed here.

| Site | Type | SOC Parameters | | Soil Physical-Chemical Parameters | | | | | Soil Fertility Parameters | | | | | |
|---|---|---|---|---|---|---|---|---|---|---|---|---|---|---|
| | | C Content g/kg | C Stock kg/m$^2$ | Bulk Density (g/cm$^3$) | Porosity (%) | Soil Moisture (%) | pH | EC (µS/cm) | Total N (g/kg) | Alkaline Hydrolyzed N(mg/kg) | Total K (g/kg) | Available K (mg/kg) | Total P (g/kg) | Available P(mg/kg) |
| Dumeng | Farmland | 7.31 | 2.29 | 1.60 | 33.17 | 6.18 | 8.35 | 107.85 | 0.63 | 45.54 | 57.94 | 56.51 | 0.35 | 5.57 |
| | Poplar | 7.68 | 2.26 | 1.51 | 38.09 | 4.56 | 8.50 | 80.86 | 0.68 | 47.16 | 57.57 | 65.33 | 0.33 | 4.85 |
| | Change (%) | 5.1 | −1.1 | −6.0 | 14.8 | −26.2 | 1.8 | −25.0 | 6.6 | 3.5 | −0.6 | 15.6 | −5.3 | −12.9 |
| Fuyu | Farmland | 8.77 | 2.55 | 1.49 | 37.87 | 16.81 | 8.48 | 110.82 | 0.91 | 60.46 | 50.08 | 75.37 | 0.19 | 2.74 |
| | Poplar | 8.89 | 2.41 | 1.40 | 40.46 | 16.42 | 8.49 | 133.11 | 1.01 | 65.12 | 51.37 | 84.51 | 0.51 | 3.25 |
| | Change (%) | 1.4 | −5.6 | −5.8 | 6.8 | −2.3 | 0.1 | 20.1 | 10.4 | 7.7 | 2.6 | 12.1 | 163.8 | 18.7 |
| Lanling | Farmland | 10.47 | 3.02 | 1.47 | 39.96 | 11.32 | 7.56 | 107.13 | 1.01 | 75.53 | 47.97 | 71.80 | 0.81 | 6.91 |
| | Poplar | 10.84 | 3.10 | 1.44 | 40.37 | 10.31 | 7.57 | 99.30 | 0.99 | 62.67 | 55.52 | 82.29 | 0.51 | 5.60 |
| | Change (%) | 3.6 | 2.7 | −1.9 | 1.0 | −8.8 | 0.1 | −7.3 | −2.1 | −17.09 | 15.7 | 14.6 | −37.0 | −18.9 |
| Mingshui | Farmland | 15.06 | 4.33 | 1.45 | 37.70 | 17.01 | 7.33 | 77.01 | 1.26 | 77.50 | 56.32 | 76.73 | 0.32 | 11.00 |
| | Poplar | 14.85 | 3.88 | 1.33 | 42.21 | 16.18 | 7.45 | 82.91 | 1.19 | 86.39 | 48.64 | 95.35 | 0.65 | 8.53 |
| | Change (%) | −1.4 | −10.4 | −8.8 | 11.9 | −4.9 | 1.7 | 7.7 | −5.6 | 11.5 | −13.6 | 24.3 | 106.5 | −22.4 |
| Zhaodong | Farmland | 10.56 | 2.93 | 1.40 | 44.28 | 13.74 | 8.44 | 151.43 | 1.03 | 59.28 | 42.64 | 47.82 | 0.15 | 3.39 |
| | Poplar | 11.59 | 3.16 | 1.39 | 40.35 | 12.90 | 8.53 | 120.48 | 1.07 | 57.15 | 51.04 | 58.45 | 0.46 | 3.72 |
| | Change (%) | 9.8 | 7.6 | −0.7 | −8.9 | −6.1 | 1.1 | −20.4 | 3.9 | −3.6 | 19.7 | 22.2 | 201.5 | 9.8 |
| Zhaodzhou | Farmland | 8.49 | 2.37 | 1.41 | 40.20 | 9.60 | 8.57 | 118.26 | 0.84 | 57.93 | 49.61 | 45.84 | 0.22 | 7.48 |
| | Poplar | 8.09 | 2.20 | 1.38 | 42.93 | 8.76 | 8.57 | 138.85 | 0.84 | 49.91 | 53.92 | 44.77 | 0.39 | 8.70 |
| | Change (%) | −4.8 | −7.1 | −2.5 | 6.8 | −8.8 | 0.0 | 17.4 | 0.0 | −13.8 | 8.7 | −2.3 | 76.1 | 16.3 |
| Change (%) | Average | 1.5 | −3.4 | −4.3 | 4.8 | −7.4 | 0.9 | −2.5 | 2.1 | −2.7 | 4.4 | 15.1 | 40.0 | −4.6 |
| | Range | −4.8%~ 9.8% | −10.4%~ 7.6% | −8.8%~ −0.7% | −8.9%~ 14.8% | −26.2%~ −2.3% | 0%~ 1.8% | −25%~ 20% | −5.6%~ 10.4% | −17%~ 11.5% | −13.6%~ 19.7% | −2.3%~ 24.3% | −37%~ 201.5% | −22.4%~ 18.7% |
| *p*-level | | 0.000 | 0.000 | 0.000 | 0.000 | 0.000 | 0.000 | 0.000 | 0.000 | 0.000 | 0.000 | 0.000 | 0.000 | 0.000 |

In the case of soil fertility parameters, shelterbelt-induced changes in N, P, K, and their available forms were different at different regions ($p$ = 0.000) (Table 3). Total N accumulation was found at Dumeng, Fuyu, and Zhaodong, and a large depletion was at Lanling and Mingshui. Alkaline hydrolyzed N of shelterbelts compared with farmland varied largely among sites; 3.5%, 7.7%, and 11.5% increases were observed in Dumeng, Fuyu, and Mingshui. The 17.1%, 3.6%, and 3.8% decreases were discovered in Lanling, Zhaodong, and Zhaozhou. Accumulations in soil total K and available K concentration of shelterbelts compared to farmland, respectively, were 4.4% and 15.1% for the overall data average (Table 2). Shelterbelt-induced changes in total K were found from −13.6% to 19.7% among six sites, and the changes in available K were from −2.3% to 24.3% among six sites, respectively. A 40.0% increase in total P in shelterbelts was found for the overall data average, and different changes were among the six sites largely ($p$ = 0.000). Peak accumulation was in Zhaozhou (201.5% increase), while the largest depletion was found in Lanling (37% decrease). Shelterbelt-induced changes in soil-available P varied from 22.4% depletion in Mingshui to 18.7% accumulation in Fuyu. An accumulation of available P was observed in Fuyu, Zhaodong, and Zhaozhou, while a depletion was found in Dumeng, Lanling, and Mingshui (Table 3).

*3.4. Shelterbelt-Induced Soil Changes Dependence on Soil Depth: Parameters, Size, and Pattern of the Changes*

Significant interactions were found between land use type and soil depth (type*depth) in 14 soil parameters (Table 1, $p$ = 0.000), and Table 4 listed the specific size and pattern of these interactions.

Differences (shelterbelt plantations-farmland) in SOC concentration were 6.5%–7.8% increases in 20–60 cm layers, while a slight decrease was found in the top layer (0–20 cm, −1.0%) and 60–100 cm layers (−3.8% to −0.5%). Similar depth-related variations in SOC stock were found in 5 soil layers, i.e., accumulation in 20–60 cm soil but depletion in other layers was observed (Table 4).

Shelterbelt-induced changes in soil physical-chemical properties varied between the surface layer (0–20 cm) and other layers (Table 4). For example, a 0.3% increase in soil-specific gravity of shelterbelt plantations was found in the surface layer. A 0.3%–3.6% decrease was discovered in other layers. A 2.9% increase in soil moisture of shelterbelt plantations compared to farmland was found in the surface layer; 6.7%–13.7% decreases were discovered in deeper layers. pH in shelterbelt plantations was a 3.1% increase in the surface layer, and slight differences were found in other layers. EC in shelterbelt plantations was 34.3% lower in the surface layer when compared with farmland, while increases in 20–80 cm layers were observed (Table 4).

Soil fertility changes from shelterbelt construction were also different from different soil depths (Table 4). Soil total P increased in all five soil layers following shelterbelt plantations construction. In contrast, the extent of increase was relatively small in the surface and bottom layers, while much more significant increases (4–5 times) were observed in 20–60 cm layers. A 7.9% and 17% decrease in soil total K in shelterbelt plantations compared to farmland were observed in surface (0–20 cm) and bottom (80–100 cm) layers, while 3.4–26.4% increases were found in 20–80 cm layers. Differences (shelterbelt plantations compared to farmland) in available P and K in surface (0–20 cm) differed from other layers. For example, an 18.2% increase was observed in available P of shelterbelt plantations in the surface layer, while a 4%–27.6% decline in other layers was generally observed. A 62% increase was observed in available K of shelterbelt plantations in the surface layer, while a 31.5% decrease was found in the 20–40 cm layer. In more deep layers >60 cm, 12.4% to 21.7% increases were observed in shelterbelt plantations (Table 4).

**Table 4.** Shelterbelt construction and sampling depth significantly affected 14 soil parameters (marked type*depth interaction, $p < 0.05$) and the magnitudes and pattern of the influences. The non-significant factors ($p > 0.05$) were not listed here.

| Depth (cm) | Type | SOC Parameters | | | Soil Physical-Chemical Parameters | | | | | | Soil Fertility Parameters | | | | |
|---|---|---|---|---|---|---|---|---|---|---|---|---|---|---|---|
| | | C Content (g/kg) | C Stock (kg/m²) | Specific Gravity | Bulk Density (g/cm³) | Porosity (%) | Soil Moisture (%) | pH | EC (μS/cm) | Total N (g/kg) | Alkaline Hydrolyzed N (mg/kg) | Total K (g/kg) | Available K (mg/kg) | Total P (g/kg) | Available P (mg/kg) |
| 0–20 | Farmland | 17.36 | 4.89 | 2.54 | 1.42 | 42.29 | 12.65 | 7.84 | 161.26 | 1.42 | 107.60 | 44.31 | 83.49 | 0.75 | 8.32 |
| | Poplar | 17.19 | 4.65 | 2.55 | 1.37 | 45.31 | 13.02 | 8.08 | 106.02 | 1.40 | 108.53 | 40.80 | 135.25 | 0.87 | 9.83 |
| | Change (%) | −1.0 | −4.9 | 0.3 | −3.8 | 7.1 | 2.9 | 3.1 | −34.3 | −1.4 | 0.9 | −7.9 | 62.0 | 16.3 | 18.2 |
| 20–40 | Farmland | 12.52 | 3.60 | 2.48 | 1.46 | 39.96 | 14.36 | 7.97 | 108.53 | 1.26 | 80.28 | 52.09 | 63.43 | 0.25 | 5.93 |
| | Poplar | 13.50 | 3.68 | 2.39 | 1.39 | 41.56 | 12.39 | 8.01 | 121.00 | 1.38 | 76.47 | 53.84 | 43.46 | 0.47 | 4.30 |
| | Change (%) | 7.8 | 2.1 | −3.6 | −4.9 | 4.0 | −13.7 | 0.5 | 11.5 | 9.3 | −4.7 | 3.4 | −31.5 | 86.9 | −27.6 |
| 40–60 | Farmland | 8.66 | 2.51 | 2.41 | 1.47 | 37.71 | 12.75 | 8.23 | 96.46 | 0.94 | 64.39 | 44.88 | 61.94 | 0.20 | 5.60 |
| | Poplar | 9.22 | 2.54 | 2.34 | 1.41 | 39.30 | 11.46 | 8.13 | 114.59 | 0.87 | 54.55 | 54.52 | 57.83 | 0.42 | 5.08 |
| | Change (%) | 6.5 | 1.1 | −2.8 | −3.9 | 4.2 | −10.1 | −1.2 | 18.8 | **−7.7** | −15.3 | 21.5 | −6.6 | 111.8 | −9.3 |
| 60–80 | Farmland | 6.96 | 2.06 | 2.41 | 1.50 | 36.15 | 11.30 | 8.26 | 99.45 | 0.63 | 28.08 | 51.86 | 50.11 | 0.24 | 5.55 |
| | Poplar | 6.70 | 1.89 | 2.40 | 1.43 | 39.37 | 10.55 | 8.33 | 110.99 | 0.66 | 34.23 | 65.53 | 61.00 | 0.31 | 4.39 |
| | Change (%) | −3.8 | −8.4 | −0.3 | −4.4 | 8.9 | −6.7 | 0.8 | 11.6 | 4.3 | 21.9 | 26.4 | 21.7 | 26.5 | −20.9 |
| 80–100 | Farmland | 5.03 | 1.50 | 2.50 | 1.51 | 38.21 | 11.16 | 8.31 | 94.72 | 0.50 | 33.18 | 60.67 | 52.77 | 0.26 | 5.51 |
| | Poplar | 5.01 | 1.41 | 2.38 | 1.44 | 38.13 | 10.19 | 8.38 | 93.67 | 0.51 | 33.21 | 50.36 | 61.38 | 0.31 | 5.28 |
| | Change (%) | −0.5 | −6.1 | −4.9 | −4.6 | −0.2 | −8.6 | 0.8 | −1.1 | 3.5 | 0.1 | −17.0 | 16.3 | 20.8 | −4.0 |
| Change (%) | Average | 1.5 | −3.4 | −2.0 | −4.3 | 4.8 | −7.4 | 0.9 | −2.5 | 2.1 | −2.7 | 4.4 | 15.1 | 40.0 | −4.6 |
| | Range | −1%~ 7.8% | −8.4%~ 2.1% | −4.9%~ 0.3% | −4.9%~ −3.8% | −0.2%~ 8.9% | −13.7%~ 2.9% | −1.2%~ 3.1% | −34.3%~ 18.8% | −7.7%~ 9.3% | −15.3%~ 21.9% | −17.0%~ 26.4% | −31.5%~ 62% | 16.3%~ 111.8% | −27.6%~ 18.2% |
| *p*-level | | 0.000 | 0.000 | 0.012 | 0.000 | 0.000 | 0.000 | 0.000 | 0.000 | 0.000 | 0.000 | 0.000 | 0.000 | 0.000 | 0.000 |

### 3.5. Shelterbelt-Induced Soil Changes Dependence on Tree Growth: Parameters, Size, and Pattern of the Changes

Significant interactions between land use type and tree height (type*H) were found in soil bulk density, soil moisture, pH, and SOC concentration. The significant interactions between land use type and DBH (type*DBH) were in pH and total K (Table 1). Table 5 lists the specific size and pattern of the difference.

**Table 5.** Shelterbelt construction and tree growth stage significantly affected 5 soil parameters (marked type*height, and type*DBH interactions, $p < 0.05$) and magnitudes and patterns of the influences. The non-significant factors ($p > 0.05$) were not listed here.

| Type | Height Group | Bulk Density g/cm$^3$ | Soil Moisture % | pH | SOC g/kg | DBH Group | pH | Total K g/kg |
|---|---|---|---|---|---|---|---|---|
| Farmland | | 1.51 | 12.83 | 8.13 | 9.09 | | 8.20 | 51.93 |
| Poplar | <12 m | 1.44 | 12.69 | 8.10 | 9.08 | <15 cm | 8.37 | 51.85 |
| Change (%) | | −4.38 | −1.08 | −0.40 | −0.11 | | 2.14 | −0.14 |
| Farmland | | 1.46 | 11.33 | 8.02 | 10.28 | | 8.10 | 51.94 |
| Poplar | 12–18 m | 1.39 | 10.79 | 8.07 | 10.58 | 15–30 cm | 8.20 | 50.77 |
| Change (%) | | −4.83 | −4.78 | 0.60 | 2.88 | | 1.20 | −2.24 |
| Farmland | | 1.45 | 13.17 | 8.21 | 10.95 | | 8.06 | 48.41 |
| Poplar | >18 m | 1.39 | 11.09 | 8.39 | 11.31 | >30 cm | 7.98 | 56.40 |
| Change (%) | | −3.77 | −15.81 | 2.14 | 3.31 | | −1.01 | 16.50 |
| *p*-value | | 0.003 | 0.000 | 0.000 | 0.031 | | 0.003 | 0.045 |

Shelterbelt plantations' construction significantly reduced soil bulk density ($p = 0.000$, Table 1). Shelterbelt-induced decrease in soil bulk density was significantly different in 3 tree height groups, and a 4.38% decrease in the <12 m group, a 4.83% decrease in the 12–18 m group, a 3.77% decrease in the >18 m group (Table 5). With tree height increase, shelterbelt-induced decrease in soil moisture increased from 1.08% to 15.81% (Table 5, $p = 0.000$), indicating that higher trees tended to absorb more soil water. Higher shelterbelt plantations tended to sequestrate more SOC in soil, e.g., <12 m trees had a similar SOC concentration (0.11% decrease), while >18 m trees could sequestrate 3.31% higher SOC (Table 5).

Shelterbelts with larger DBH recovered soil total K (from a 2.24% decrease to a 16.5% increase) compared to those with smaller DBH. Shelterbelt plantations with higher heights and smaller DBH are generally accompanied by an increased pH. For example, with the increase in tree height, shelterbelt-induced soil pH changes ranged from a 0.4% decline in the <12 m group to a 2.14% increase in the >18 m group. With the increase in DBH, shelterbelt-induced pH changes ranged from a 2.14% increase in the <15 cm group to a 1.01% decline in the >30 cm group (Table 5).

### 3.6. Regression Analysis between Shelterbelt-Induced SOC Changes and Variable Soil Properties

Shelterbelt plantations construction significantly affected physical-chemical and fertility properties (Tables 1–5). Correlations between soil properties change and afforestation-induced SOC changes in concentration and storage were listed in Table 6. Significant linear relationships were found between shelterbelt-induced SOC change and the changes in bulk density, porosity, moisture, pH, and EC ($p < 0.05$). As shown in Table 6, positive correlations were found in SOC stock change and soil bulk density change (r = 0.15, $p < 0.01$, n = 360), SOC concentration change, and soil EC (r = 0.11, $p < 0.05$, n = 360). Negative correlations were found between SOC stock change and soil porosity (r = −0.13, $p < 0.05$, n = 360), SOC stock change and soil moisture (r = −0.11, $p < 0.05$, n = 360), SOC concentration change and soil pH (r = −0.11, $p < 0.05$, n = 360). Significant linear relationships were also found in soil fertility parameters, such as total N, alkaline hydrolyzed N,

and total P (Table 6). Shelterbelt-induced SOC changes were positively correlated with the changes in alkaline hydrolyzed N (r > 0.11, $p < 0.05$, n = 360). While negative relations were found in total P (r = −0.16, $p < 0.01$, n = 360). Compared to all other significant relations, the strongest relationship was found between shelterbelt-induced SOC changes and total N changes in concentration (r = 0.50, $p < 0.001$, n = 360) and stock (r = 0.51, $p < 0.001$, n = 360) (Figure 2; Table 6). By taking the linear gradient as the rate of SOC accumulation with soil N increment, one g kg$^{-1}$ of N increase in soil was accompanied by 4.4 g kg$^{-1}$ or 1.27 kg m$^{-2}$ in SOC, respectively (Figure 2a,b). The increases in total N content and stock were accompanied by a 1.07 and 1.11-fold higher SOC content and stock sequestration in 0–100 cm layers in poplar forests (Figure 2c–f).

**Table 6.** Correlations between shelterbelt-induced changes in various soil properties and SOC sequestration in concentration and storage.

| Soil Properties Change (Shelterbelt-Farmland) | Sample Number | SOC Concentration Change (Shelterbelt-Farmland) | | | SOC Stock Change (Shelterbelt-Farmland) | | |
|---|---|---|---|---|---|---|---|
| | | Linear Equation | r | *p*-Level | Linear Equation | r | *p*-Level |
| Soil physical-chemical change | | | | | | | |
| Specific gravity | 360 | Y = −0.40X + 0.13 | −0.05 | $p > 0.05$ | Y = −0.13X − 0.11 | −0.06 | $p > 0.05$ |
| Bulk density | 360 | Y = −2.96X − 0.04 | −0.09 | $p > 0.05$ | Y = 1.38X − 0.01 | 0.15 | $p < 0.01$ |
| Porosity | 360 | Y = −0.01X + 0.16 | −0.03 | $p > 0.05$ | Y = −0.01X − 0.08 | −0.13 | $p < 0.05$ |
| Soil moisture | 360 | Y = −0.07X + 0.09 | −0.06 | $p > 0.05$ | Y = −0.03X − 0.13 | −0.11 | $p < 0.05$ |
| pH | 360 | Y = −0.97X + 0.22 | −0.11 | $p < 0.05$ | Y = −0.24X − 0.09 | −0.09 | $p > 0.05$ |
| EC | 360 | Y = 0.01X + 0.16 | 0.11 | $p < 0.05$ | Y = 0.00X − 0.10 | 0.08 | $p > 0.05$ |
| Soil fertility change | | | | | | | |
| Total N | 360 | Y = 4.40X − 0.06 | 0.50 | $p < 0.001$ | Y = 1.27X + 0.02 | 0.51 | $p < 0.001$ |
| Alkaline hydrolyzed N | 360 | Y = 0.01X + 0.16 | 0.13 | $p < 0.05$ | Y = 0.01X − 0.09 | 0.11 | $p < 0.05$ |
| Total K | 360 | Y = 0.01X + 0.14 | 0.03 | $p > 0.05$ | Y = 0.02X − 0.10 | 0.09 | $p > 0.05$ |
| Available K | 360 | Y = 0.00X + 0.13 | 0.03 | $p > 0.05$ | Y = 0.00X − 0.11 | 0.03 | $p > 0.05$ |
| Total P | 360 | Y = −0.41X + 0.20 | −0.09 | $p > 0.05$ | Y = −0.70X − 0.08 | −0.16 | $p < 0.01$ |
| Available P | 360 | Y = −0.05X + 0.13 | −0.09 | $p > 0.05$ | Y = −0.02X − 0.11 | −0.04 | $p > 0.05$ |

Table 7 shows the spatial differences (site and depth) in the correlations between shelterbelt-induced N changes and SOC changes. The most evident relations were found in 40–60 cm with r values larger than 0.70. By taking the linear gradient as the rate of SOC accumulation with soil N increment, N increase in deep soils was accompanied by higher SOC accumulations. For example, SOC accumulation 0–40 cm was 3.79–4.46 g g$^{-1}$ N or 1.08–1.29 kg m$^{-2}$ for one-unit N increase, while those for deeper soils were respectively 4.56–5.65 g g$^{-1}$ N or 1.37–1.73 kg m$^{-2}$.

The different site also has different relations (Table 7). The strongest relationship was found in Fuyu (r = 0.64), while the weakest one was found in Dumeng (r = 0.30). All sites showed that the N increase in soil was accompanied by SOC accumulation. However, the site variations in the rate (linear gradient) were as large as 2.2-fold for concentration and 1.9-fold for storage (Table 7).

Figure 3 shows that farmland and shelterbelt differences in the correlations between total soil N and SOC at different soil layers. The slope value showed that the SOC changing rate with soil N changes. In surface 0–20 cm soils, the increase in total N was accompanied by 1.11-fold SOC accrual in farmlands (7.87) than in poplars (7.09). However, the increase in total N accompanied 1.01–1.67-fold higher SOC sequestration in deep soils >20 cm in poplar forests.

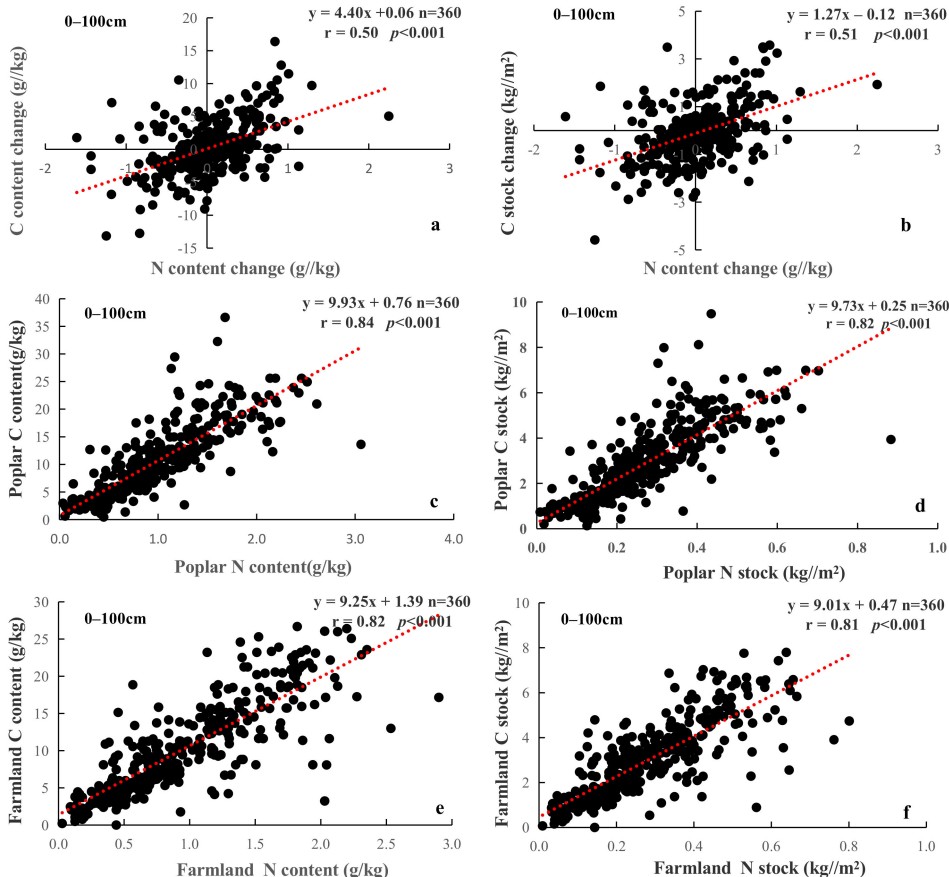

**Figure 2.** Correlations between total N content change(shelterbelt-farmland) and SOC changes (shelterbelt-farmland differences) at 0–100 cm. (**a**) in content; (**b**) in stocks. Correlations between poplar total N and poplar SOC at 0–100 cm. (**c**) in content; (**d**) in stocks. Correlations between farmland total N and farmland SOC at 0–100 cm. (**e**) in content; (**f**) in stocks.

**Table 7.** Differences in the total N-SOC sequestration potential relations in different soil layers and sites.

| | Sample Number | SOC Concentration Change (Shelterbelt-Farmland) | | | SOC Stock Change (Shelterbelt-Farmland) | | |
|---|---|---|---|---|---|---|---|
| | | Linear Equation | r | *p*-Level | Linear Equation | r | *p*-Level |
| Depth (cm) | | | | | | | |
| 0–20 | 72 | Y = 4.46X − 0.13 | 0.50 | $p < 0.001$ | Y = 1.29X − 0.03 | 0.51 | $p < 0.001$ |
| 20–40 | 72 | Y = 3.79X + 0.47 | 0.44 | $p < 0.001$ | Y = 1.08X + 0.13 | 0.44 | $p < 0.001$ |
| 40–60 | 72 | Y = 5.17X + 0.89 | 0.71 | $p < 0.001$ | Y = 1.50X + 0.26 | 0.71 | $p < 0.001$ |
| 60–80 | 72 | Y = 4.53X − 0.54 | 0.49 | $p < 0.001$ | Y = 1.37X − 0.16 | 0.50 | $p < 0.001$ |
| 80–100 | 72 | Y = 5.65X − 0.31 | 0.57 | $p < 0.001$ | Y = 1.73X − 0.10 | 0.57 | $p < 0.001$ |
| Site | | | | | | | |
| Dumeng | 60 | Y = 2.91X + 0.12 | 0.30 | $p < 0.05$ | Y = 0.99X + 0.03 | 0.40 | $p < 0.01$ |
| Fuyu | 60 | Y = 6.45X − 0.58 | 0.64 | $p < 0.001$ | Y = 1.91X − 0.18 | 0.64 | $p < 0.001$ |
| Lanling | 60 | Y = 4.40X + 0.64 | 0.61 | $p < 0.001$ | Y = 1.22X + 0.17 | 0.58 | $p < 0.001$ |
| Mingshui | 60 | Y = 4.44X + 0.04 | 0.53 | $p < 0.001$ | Y = 1.34X + 0.02 | 0.54 | $p < 0.001$ |
| Zhaodong | 60 | Y = 3.61X + 0.65 | 0.42 | $p < 0.001$ | Y = 1.02X + 0.19 | 0.42 | $p < 0.001$ |
| Zhaozhou | 60 | Y = 5.03X − 0.59 | 0.59 | $p < 0.001$ | Y = 1.37X − 0.15 | 0.59 | $p < 0.001$ |

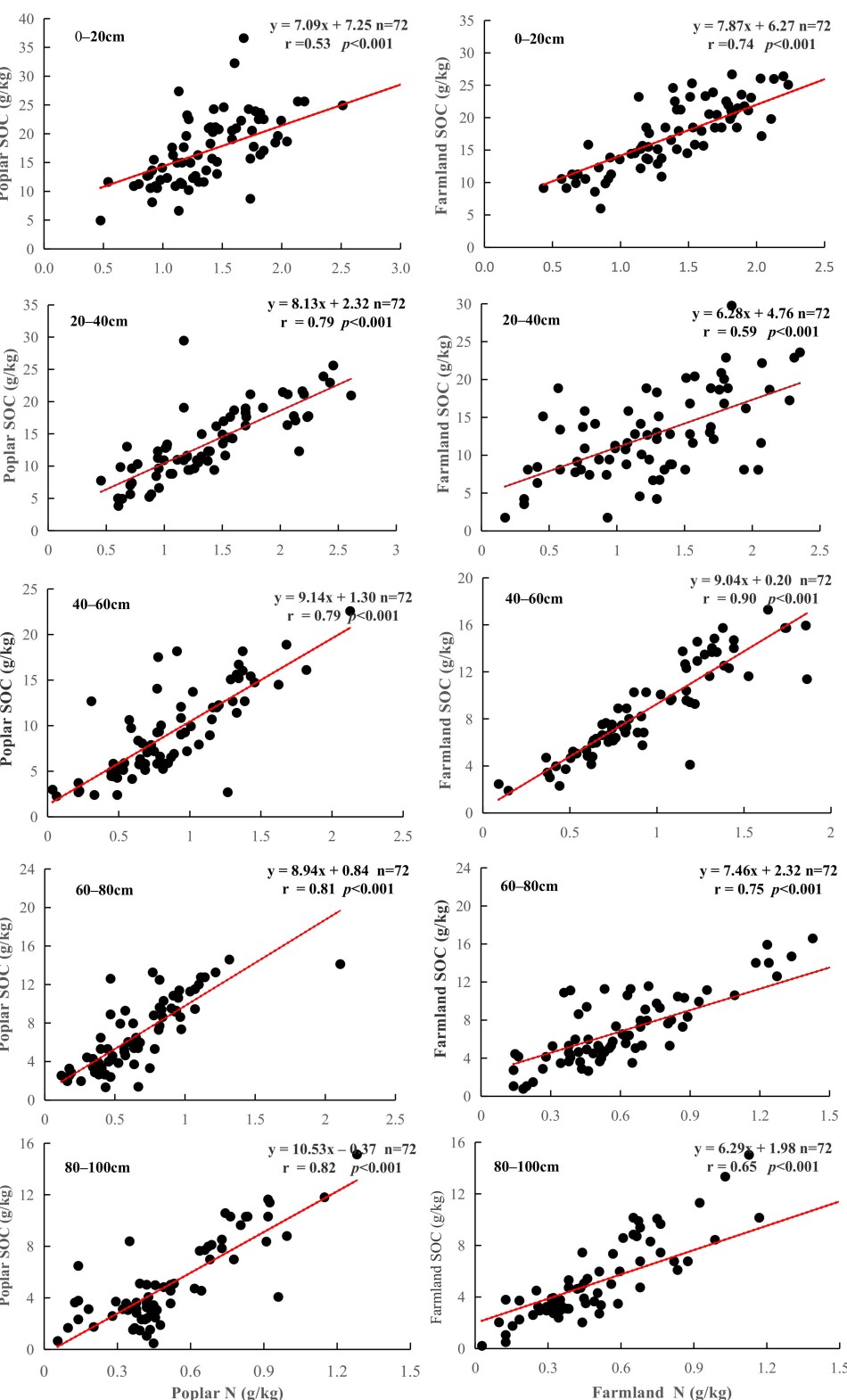

**Figure 3.** Differences in the correlations between soil total N concentration and SOC concentration in poplar and farmland at different soil depths. Left: poplar shelterbelts. Right: farmlands. From the top raw to the lowest raw of the figures are 0–20 cm, 20–40 cm, 40–60 cm, 60–80 cm, and 80–100 cm soil layer, respectively. The slopes for total N-SOC in poplar in deep soils >20 cm were 1.29, 1.01, 1.19, and 1.67-fold higher than those in farmland, while in 0–20 cm, farmland had 1.11-fold higher slope value than the poplar shelterbelt.

## 4. Discussion

At the global scale, the worldwide ecological shelterbelt engineering projects, such as the Great Plains Shelterbelt Project in the USA, the Great Plan for the Transformation of Nature in the former Soviet Union, the forestry and water conservation projects in Japan, the Green Dam Engineering Project in the five countries of North Africa and Three-North Shelterbelts in China, promoted the development of the science of shelterbelt forests [22]. Among the vast amount of forest plantations in the world, many of them planted in degraded or abandoned farmland is used as agricultural protection forests. Shelterbelt forest areas used for protecting soil and water have increased by 330 Mha in all regions of the world until 2010 and account for 8% of total forest areas. The highest proportion of shelterbelt forests is in Asia (26%), and 33% have been reported in East Asia. The shelterbelt forests in China account for the most areas (60 Mha of the total 83 Mha) [31]. About 6.67 Mha poplar plantations are widely distributed in China. As an excellent example of shelterbelt forests, a systematic study in China will benefit the scientific understanding of their function in soil functional maintenance [22]. Compared with other references, the following were discussed to highlight the function of shelterbelt forests on soil physics: fertility, water, and SOC sequestrations.

### 4.1. Shelterbelt-Afforestation Improves Soil Physical Properties and Soil Fertility with Higher Water Consumption

Farmland shelterbelts construction significantly improved soil physics with soil bulk density decreases (4.3% decrease) and soil porosity increases (4.8% rise) (Table 2, $p < 0.01$). There were consistent results in different locations and different soil depths (Tables 3 and 4). Our finding indicated that farmland shelterbelts construction could make soil looser and more porous, and other authors also reported similar results. Marta and Halina [32] found that the total soil porosity in the whole horizon was 1.08 and 1.12 times higher in the young and older studied afforested soils than in the respective arable soils. Soil bulk density decreased by 5.7 mg cm$^{-3}$ yr$^{-1}$ in the 0–20 cm soil layer in returning farmland to larch plantations in northeastern China [5]. Soil bulk density and porosity are important physical parameters for gas penetration, water transportation, and soil nutrient storage [33]. Long-term farmland cultivation has seriously degraded black soil in northeastern China, and one crucial aspect is soil physics degradation [19]. Our result manifested that shelterbelt afforestation can enormously improve soil physics and hints at a possible way for local soil improvement, such as returning degraded farmland to forests; this policy has been implemented in China for years [5].

High water consumption should be another feature of the poplar shelterbelts construction, and our results support this (Tables 2–4). Shelterbelts produced an overall 7.4% decrease in soil moisture (Table 2, $p = 0.000$). This reduction differed from different regions (Table 3), different soil depths (Table 4), and different tree growths (Table 5). In the Loess Plateau of China, Yang, Wei [34] have manifested that forestland reduction in soil moisture, followed by native grassland, and traditional farmland had the highest value. In artificial afforestation, leaf interception, and root uptake can decrease soil moisture [35], which leads to consuming large amounts of water. Qiu, Pan [36] found that the water consumption of 4 fast-growing trees was *Catalpa bungei* > *Populus tomentosa* > *Salix sp.* > *Eucalyptus urophylla* × *Eucalyptus grandis*. Previous studies also found that the soil moisture differed significantly between introduced arbor vegetation and traditional farmland [37,38]. In combination with our results, we concluded that higher water consumption in poplar shelterbelt plantations could intensify the degree of drought in this region. This region distributed a large area of saline-alkali land with an average precipitation of 400–500 mm [39]. Some measures were used to solve this bad-effect problem, such as digging a root-cutting ditch to hinder the farmland's root invasion. Possible measures have been proposed for counteracting the over-water consumption from plantation forests [40], i.e., the proportion of native forests plays a key role in the regulation and reduction of water use. Therefore, a system of mosaic management may be able to stabilize water flow across plantation landscapes. A study in this region has found that some local species, such as Ulmus, could

improve soil properties [41], and more diversified species could increase water interception of whole forests (Jin Lixin, unpublished data).

Beyond water consumption and soil physical properties improvement, farmland shelterbelts construction significantly increased the soil total P (40%), total K (4.4%), and available K (15.1%) (Table 2). This pattern differed in different growth stages (−2.24% to 16.5%) (Table 5). Gao and Huang [42] observed that compared with farmland, available K (71.13%) and available P (14.17%) contents significantly increased in the 0–20 cm soil layer following the construction of the "Three-North Shelter Forest" in Northwestern China. The soil fertility of P and K could be rehabilitated after returning farmland to the forest in different areas of the world with different trees [43,44]. One reason should be the nutrient absorption differences between crops and poplar trees. Respectively, P and K in poplar trees were 1.4 and 8.9 g kg$^{-1}$ [45]. As the two main crops in the local region, the P and K concentration in maize was 2.6 and 17.0 g kg$^{-1}$ [46], respectively, and their respective engagement in soybean was 5.9 and 17.2 g kg$^{-1}$ [47]. Given the similar productivity of shelterbelt poplar and crops, crops could consume 1.9–4.2-fold higher P and 1.92–1.94-fold higher K. These large differences in consumption of P and K, should contribute to the soil nutrient recovery after shelterbelts construction in farmland. In the case of farmland fertilization practice, more P chemical fertilizer and N (the favorite fertilizer for local people) should be applied to secure soil nutrient supply for crop productivity.

*4.2. Complexity in Changes of SOC after Building Shelterbelts on Farmland: Deep Soil Importance, Tree Growth Status, and Site Differences*

As one of our main findings, shelterbelts forest construction on farmland did not result in an overall change in SOC (all six sites throughout 1 m soil profile) ($p > 0.05$) (Table 1). Some sites were SOC sinks after afforestation, such as Zhaodong, Dumeng, Lanling, and Fuyu. The other sites (Mingshui and Zhaozhou) were carbon sources compared with farmland (Table 3). Some soil depths in shelterbelt plantations, such as 20–40 cm and 40–60 cm layers, were most likely SOC sinks, while SOC concentration and storage in other layers (0–20 cm and 60–100 cm) decreased with afforestation compared with neighboring farmland (Table 4). Shelterbelt plantations with higher tree height tended to sequestrate more SOC in soil (Table 5). Our finding indicates that SOC sequestration after shelterbelts construction is more complex than in previous reports [6]. At least three aspects should be fully considered, i.e., deep soil importance, tree growth status, and site differences.

SOC accumulation and performance studies in deep soils have been highlighted in recent studies [10,48]. Additionally, our study highlights that surface soil, subsoil, and even soil deeper than root reaches should be fully considered in studying shelterbelt forest influences on soil C sequestration (Table 4). According to our previous survey for 1 m depth, 95% poplar root system was distributed in 0–60 cm soil layer, especially in 20–40 cm (57%) [14]. Our study found that the SOC stock 6.5%–7.8% increase in the 20–60 cm after the conversion of cropland into poplar shelterbelts (Table 4). Differed from 20 to 60 cm soil layers, 3.8% and 0.5% decreases in SOC were discovered in deeper layers (60–100 cm, Table 4), which is deeper than root reaches. These contrary changes in the vertical profile (0–100 cm) resulted in unremarkable SOC accumulation in the overall soil profile (Table 1). Chang, Fu [49] discovered that compared with former cropland, the SOC stock of locust forest significantly increased in subsoil (30–60 cm soil layer) during the 30 years. Hooker and Compton [50] also found that the SOC in the top 20 cm layer was found not to differ over a century following the establishment of a white pine forest on former arable land in the USA by their chronosequence study; however, the SOC could linearly accumulate at a significant rate in 20–70 cm soil layer. In larch forest plantations, SOM changing rates at 0–20 cm soil layer was 262.1 g kg$^{-1}$ year$^{-1}$, and contrary tendencies in deeper soils resulted in no significant changes in the overall 80-cm soil profile [51]. Furthermore, subsoil deeper than most roots reaches with minimal fresh C transportation (in this paper, soil over 60 cm in depth) should also be highlighted [52], and the decomposition rate of SOC in the subsoils lacking roots was much lower than the topsoils [48]. The lack of energy input

from plant roots due to the decrease in root density enhances the persistence of SOC in subsoil [53]. The root exudate inputs can stimulate the decomposition of SOC by priming soil microbial activity in deep soil [54]. The SOC mineralization may stimulate the loss of a deeper SOC pool [55,56], which has been shown in this paper by the SOC decrease in 60–100 cm (Table 4). Similar to these previous reports, our results indicate that subsoil should be considered in estimating SOC sequestration following shelterbelts construction.

The growth status of plantation forests is an essential factor influencing SOC stocks. Tree age was a credible parameter to describe SOC changes in previous research; however, it is difficult for popular trees owing to the unrecognizable tree rings (a typical diffuse-porous wood) [57]. Significant linear correlations between tree body size (DBH: diameter at breast height and tree height) and tree age were observed ($p < 0.001$) [58]. Thus, body sizes (instead of tree age) were used for studying the effect of tree growth status on SOC [59]. A significantly positive correlation ($p < 0.01$) was also found between total SOC storage and mean DBH of coniferous forests in Shennongjia nature reserve in China [60]. These conclusions agreed with our observation, i.e., compared with young plantations, mature shelterbelts with higher tree height tended to sequestrate more SOC in soil (from a 0.11% decrease to a 3.31% increase) (Table 5).

Site variations in SOC storage have been reported in plantation forests in many previous studies, and our study confirmed the significantly different SOC sequestration in 6 sampling regions (Tables 1 and 3). Beyond the above-mentioned tree growth, soil physical-chemical properties and fertility in other locations may also control the SOC sequestration ability [8,61]. Variable physical-chemical parameters, soil nutrients, and their available forms were concurrently measured in shelterbelt plantations and neighbor farmland in this paper, which makes it possible to check which factors possibly determine the SOC sequestration differences between farmland and shelterbelt plantations. Hopefully, this kind of discrimination of limiting factors for SOC sequestration in northeastern China will favor the management of the local ecosystem, and the next section will discuss this.

*4.3. Significant Correlation betweenSOC and Total N*

We used correlations between soil parameters (differences between shelterbelts and farmlands) and SOC sequestration to analyze the relationship between SOC and soil physical-chemical properties in this paper. Our results manifested that SOC sequestration potential was correlated with the changes in bulk density, porosity, pH, EC, total N, total P, and alkaline hydrolyzed N. In contrast, the strongest relationship was observed between total N and SOC (Tables 6 and 7, Figures 2 and 3). Similar to our study, a meta-analysis revealed that a significant relationship was observed between the rates of SOC and N stock changes in the organic layer ($r^2 = 0.83$, $p < 0.001$, n = 41) and the mineral layer ($r^2 = 0.66$, $p < 0.001$, n = 203) [62]. The N dynamics were the key factors affecting terrestrial carbon sequestration [63]. The increase in C stocks must be matched with enough N in the ecosystem. Otherwise, terrestrial C sequestration will be downgraded and will not be sustainable in the long term [64,65]. According to our results, one g kg$^{-1}$ of N increase in soil was accompanied by 4.4 g kg$^{-1}$ or 1.27 kg m$^{-2}$ SOC, respectively (Figure 2), indicating that N is important in maintaining the accumulation in forest C pool, both in biomass and soil [62,66].

Although shelterbelts construction could improve soil physics with soil bulk density decreases and soil porosity increases, recover soil fertility with P, K increases (Table 2), the soil N supply difference between farmland and shelterbelts was closely related to SOC sequestration. China is a large agricultural country with 15.89 Mha farmland, where the N fertilizer (31.79–32.95 million tons in 2010) was often applied in farming practices in China [67]. Combined with our findings, these farmland fertilizing practices will likely favor the SOC accumulations in farmland.

The close relations between N and SOC differences have some implications for af-forestation management. As one economic measure, the introduction of some plants with biological N fixing ability in shelterbelt plantations, as well as proper rotation in farm-

land, may favor SOC accumulations. Corresponding measures have been implemented in subtropical China. Introducing $N_2$-fixing tree species (*Acacia mangium*) into Eucalyptus plantations showed a noticeable increase in SOC and C-acquiring enzyme activities [68]. Further research indicated that $N_2$-fixing species increased soil carbon storage and recalcitrant carbon composition in Eucalyptus plantations by regulating the soil microbial community's function and extracellular enzyme activities [69]. In the case of shelterbelt plantations, conversion from pure poplar plantation to mixing plantation with N fixing ability will favor tree growth and SOC sequestration underground in northeast China. Some N-fixing local species, such as *Caragana sibirica*, *Alnus sibirica*, *Alnus fruticosa*, *Amorpha fruticosa*, *Lespedeza bicolor*, *Gleditsia japonica* and *Maackia amurensis*, may be selected as associated candidate plants. In the case of farmland, different legume species, including soybean, mung bean, kidney beans, forage grass of *Medicago sativa* and *Melilotus albus*, and traditional Chinese medicine of *Glycyrrhiza uralensis* and *Astragalus membranaceus,* etc. rotation with the popular crop of maize will favor SOC sequestration and soil improvements.

## 5. Conclusions

Through analyzing 720 paired soil samples from poplar shelterbelts and farmland in Songnen Plain in northeastern China, we concluded the following: (1) poplar shelterbelts construction on farmland significantly decreased soil bulk density and soil moisture, increased soil porosity and recovered soil nutrients of total P, total K, and available K. (2) For overall pooled data (different sites and full 1 m soil profile), shelterbelts afforestation in degraded farmland did not markedly affect SOC in concentration and stock ($p > 0.05$), while this SOC sequestration significantly interacted with locations, soil depths, and tree growth stages. (3) Significant correlations were found between SOC sequestration and the changes in soil bulk density, soil porosity, soil moisture, pH, EC, total N, total P, and alkaline hydrolyzed N. At the same time, the most significant positive linear correlation was found between N change and C change ($p < 0.001$). Our results suggested introducing N-fixing species in shelterbelts and farmlands can benefit C sequestration underground in local soil management, and multiple soil properties should be used to evaluate shelterbelt soil impacts.

**Author Contributions:** Conceptualization, Y.W. and W.W.; methodology, Q.W.; software, Z.Z.; validation, Y.Y., H.W. and W.W.; formal analysis, Y.W.; investigation, W.W.; resources, Y.Y.; data curation, Q.W.; writing—original draft preparation, Y.W.; writing—review and editing, Y.W.; visualization, H.W.; supervision, W.W.; project administration, W.W.; funding acquisition, W.W. All authors have read and agreed to the published version of the manuscript.

**Funding:** This study was supported financially by China's National Foundation of Natural Sciences (41877324, 41730641), Key Laboratory of Ecology and Management on Forest Fire in Higher Education institutions of Guizhou Province and Scientific Research Platform Project of Education Department of Guizhou Province (QJJ [2022] No. 051), Jilin Provincial Key Project (20200503001SF).

**Data Availability Statement:** No new data were created.

**Conflicts of Interest:** The authors declare no conflict of interest.

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
