# Peer review of "Farmland Shelterbelt Changes in Soil Properties: Soil Depth-Location Dependency and General Pattern in Songnen Plain, Northeastern China"

_forests, doi:10.3390/f14030584_

Round 1
Reviewer 1 Report (Previous Reviewer 1)
The authors complied with the recommendations required by me. The manuscript is well written and presented. It can therefore be published in present form.
Author Response
Thanks for your valuable advice.
Reviewer 2 Report (Previous Reviewer 2)
Reviewer
MDPI – Forests
Manuscript Number: forests-2287939
Title: «Farmland shelterbelt changes in soil properties: soil depth-location dependency and general pattern in Songnen Plain, north-eastern China».
The article has been significantly corrected in comparison with the previous version.
There is a small remark.
Recommended for publication.
line 498: Site variations in SOC storage have been reported in plantation forests in many pre-496 vious studies, and our study confirmed the significantly different SOC sequestration in 6497 sampling regions (p=0.000, Tables 1, 3).
What did you mean by the designation of statistical significance in brackets?
Author Response
MDPI – ForestsManuscript Number: forests-2287939Title: «Farmland shelterbelt changes in soil properties: soil depth-location dependency and general pattern in Songnen Plain, north-eastern China».The article has been significantly corrected in comparison with the previous version.There is a small remark.Recommended for publication.line 498: Site variations in SOC storage have been reported in plantation forests in many pre-496 vious studies, and our study confirmed the significantly different SOC sequestration in 6497 sampling regions (p=0.000, Tables 1, 3).What did you mean by the designation of statistical significance in brackets?
Response: Thanks for your valuable advice. “p=0.000” in bracket has been deleted in page 16 line 288.

This manuscript is a resubmission of an earlier submission. The following is a list of the peer review reports and author responses from that submission.
Round 1
Reviewer 1 Report
A report for: forests-2198478 Poplar shelterbelts improved multiple soil properties in 1m soil profiles at six regions of Songnen Plain, northeastern China
It is an interesting paper in wich authors hypothesize that poplar shelterbelts construction can improve soil multiple properties for recovery of soil fertility of the 1m soil profile, and they strongly regulated the afforestation-induced SOC sequestration in different regions and forest age groups. Despite the merits of this study, the manuscript still needs thorough editing before it can be considered for publication in international journals.
- Title doesn't seem rigorous (1 m).
Keywords: Poplar shelterbelts; soil properties; soil fertility; carbon sequestration; multivariate analysis of variance; farmlands. Please do not repeat the same words as keywords that you have already mentioned in the title.
-Line 40 …properties of carbon, water, nutrients and physics. This classification seems inappropriate.
-The objetives: the variation of SOC, soil fertility, and soil physical-chemical properties in different soil depths of shelterbelts and farmland to explore the soil improvements and limiting factors controlling SOC sequestration in this region. If the objectives are SOC sequestration, then the title does not seem appropriate.
-Line 136. Soil samples were collected from five soil 136 layers (0-20 cm, 20-40 cm, 40-60 cm, 60-80 cm, and 80-100 cm). Please explain the reasons for sampling at different depths, other than genetic horizons.
-Line 150. an acidity meter. I think it is more appropriate pH meter.
- Please follow the journal rules, specially reference section.
- My main concerns are related to the extensive and somtimes difficult to understand some sentences. I propose to the authors to be more specific, explanatory and simplified in order to be easily understandable from the readers.
Reviewer 2 Report
Reviewer
MDPI – Forests
Manuscript Number: forests-2198478
Title: «Poplar shelterbelts improved multiple soil properties in 1m soil profiles at six regions of Songnen Plain, northeastern China ».
The title of the article should be made more concise, without mentioning the elements of the methodology. Now the name sounds like some kind of task.
line 40. The first highlight is an element of methodology, not an achievement that you have achieved and affects fundamental science.
line 106. There is a scientific hypothesis, but there is no research goal.
line 148 The formula of soil porosity must be formatted correctly: from a new paragraph with alignment to the width of the sheet.
table 1 The "p-value" indicator cannot be equal to zero - 0.000. It is likely that there are values of 0.0001.
Table 2, 3, 4, 5. The title of this table needs to be reformulated. Name must not start with a number.
line 530 Figure 3. The figure should appear after the first mention in the text.
Round 2
Reviewer 1 Report
The proposed new title does not seem reasonable either. I beg to change it. A possible title could be this: Farmland shelterbelt changes in soil properties: soil depth-location dependency and general pattern in Songnen Plain, northeastern China”. The new first highlight that they propose is not reasonable either. Please change it for another.
